# Effective Text-to-Image alignment with Quality Aware Pair Ranking

## Abstract

Fine-tuning techniques such as Reinforcement Learning with Human Feedback (RLHF) and Direct Preference Optimization (DPO) allow us to steer Large Language Models (LLMs) to be align better with human preferences. Alignment is equally important in text-to-image generation. Recent adoption of DPO, specifically Diffusion-DPO, for Text-to-Image (T2I) diffusion models has proven to work effectively in improving visual appeal and prompt-image alignment. The mentioned works fine-tune on Pick-a-Pic dataset, consisting of approximately one million image preference pairs, collected via crowdsourcing at scale. However, do all preference pairs contribute equally to alignment fine-tuning? Preferences can be subjective at times and may not always translate into effectively aligning the model. In this work, we investigate the above-mentioned question. We develop a quality metric to rank image preference pairs and achieve effective Diffusion-DPO-based alignment fine-tuning.We show that the SD-1.5 and SDXL models fine-tuned using the top 5.33% of the data perform better both quantitatively and qualitatively than the models fine-tuned on the full dataset. The code is available at this link.

## 1 Introduction

Currently, diffusion-based Text-to-Image (T2I) (Rombach et al., 2021; Podell et al., 2023; Chen et al., 2023; 2024) models are state-of-the-art in image generation. These models are trained in a single stage on a large-scale dataset of images scraped from the internet, enabling them to have huge knowledge. However, their outputs often fail to align with human preferences, as they are not explicitly optimized for this purpose. In contrast, Large Language Models (LLMs) undergo training in two distinct stages: the first stage involves pre-training on large web-scale datasets, while the second stage uses Supervised Fine-tuning (SFT) and Reinforcement Learning based on Human Feedback (RLHF) to align outputs with human preferences. While significant progress has been made in alignment fine-tuning for LLMs, aligning T2I outputs with human preferences remains a difficult challenge.

Recent works have begun exploring how to better align T2I models with human preferences. These approaches can be broadly classified into two broad categories – they either use a reward model trained on human preference data to guide the T2I model, or they directly fine-tune the T2I model on pairwise preference data. Reinforcement Learning (RL) based approaches like Alignprop(Prabhudesai et al., 2023), ImageReward(Xu et al., 2023), DDPO(Black et al., 2023) do not scale well to large datasets and are highly prone to problems like overfitting and mode collapse. Additionally, training good reward models and using them to fine-tune diffusion models introduces significant operational challenges, as it adds a lot of computational overhead.

To address this gap in diffusion model alignment, approaches like Diffusion-DPO(Wallace et al., 2023a) have emerged, reformulating the loss function to completely remove the reward model and directly fine-tune on pairwise image preference data, which solves the problems of traditional RL-based approaches. In recent works, more preference alignment approaches like Diffusion-KTO(Li et al., 2024) and IPO(Azar et al., 2023) have emerged, building on Diffusion-DPO to further improve diffusion model alignment. However, all of these approaches share a common drawback: they either require pairwise preference data or a label of "good" or "bad" for each image. These labels, collected from human based annotators, can be noisy as preference is subjective. Additionally, these labels do

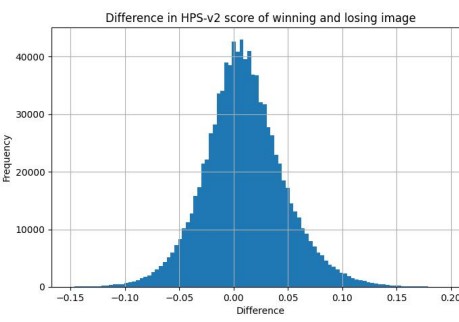 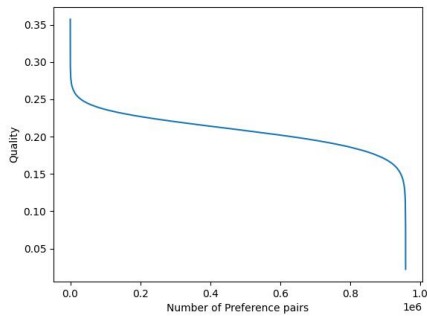

(a) Difference in HPSv2 scores (which can be viewed as probability of being preferred) of the winning image and the losing image. This wide distribution suggests that not all winning samples are equally dominant, and not all losing samples are equally inferior.

(b) Quality metric plotted in a sorted order for preference pairs in the Pick-a-Pic dataset.

Figure 1: Left - plot of difference in HPSv2 sores for Pick-a-Pic train dataset, Right - plot of quality metric on Y-axis with the sorted dataset index on X-axis.

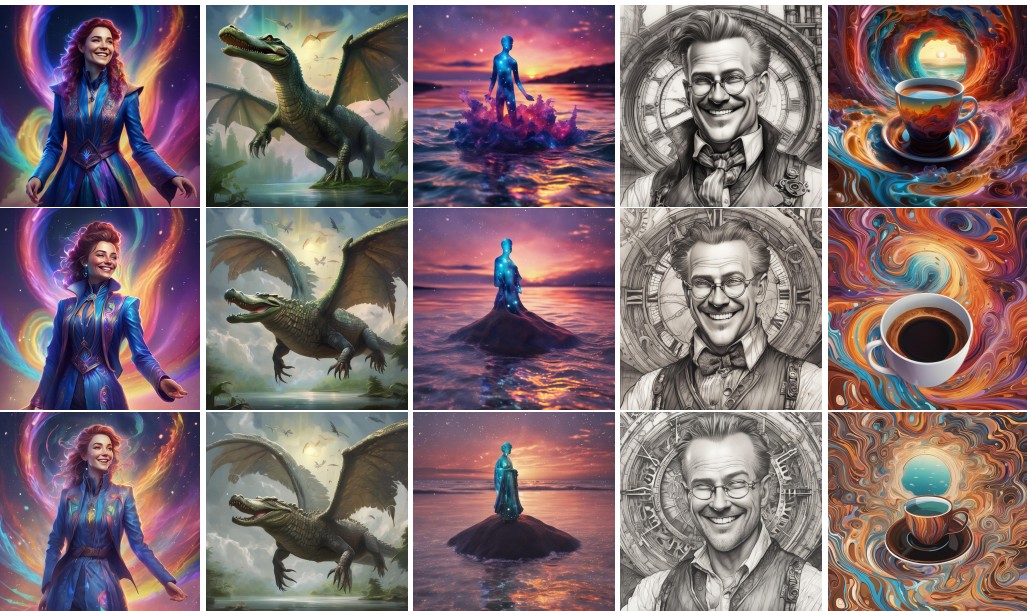

Figure 2: Top to Bottom: *SDXL-DPO-QSD, SDXL-DPO, SDXL*

Prompts: *(1) A smiling beautiful sorceress wearing a high necked blue suit surrounded by swirling rainbow aurora, hyper-realistic, cinematic, post-production (2) Concept art of a mythical sky alligator with wings, nature documentary (3) A galaxy-colored figurine is floating over the sea at sunset, photorealistic (4) close up headshot, steampunk middle-aged man, slick hair big grin in front of gigantic clocktower, pencil sketch (5) A swirling, multicolored portal emerges from the depths of an ocean of coffee, with waves of the rich liquid gently rippling outward. The portal engulfs a coffee cup, which serves as a gateway to a fantastical dimension. The surrounding digital art landscape reflects the colors of the portal, creating an alluring scene of endless possibilities.*

not capture the "strength" of the preference pair and treat each pairwise sample as equally important, which we show in Figure 1a is a huge flaw. In the graph, we plot the difference of the HPSv2(Wu et al., 2023) scores of winning images and the losing images for the Pick-a-Pic train set. As we can observe not all samples are equal and in fact follow a normal distribution with mean around 0 with some samples even having higher scores for negative samples. We believe that samples where the

winning and losing images have similar or inverse AI preference ratings negatively impact the model during pairwise preference fine-tuning by sending conflicting signals. For instance, pairs focusing on individual qualities like prompt adherence or image aesthetics might steer the model in different directions, making the learning sub-optimal, while fine-tuning on the pairs that are consistent across all qualities would result in a better model.

To address these shortcomings, we propose our novel approach — **Effective Text-to-Image Alignment with Quality Aware Pair Ranking**. Specifically, we introduce a quality metric to assess the quality of a pair of images and the corresponding prompt as a fine-tuning sample. We use a carefully devised metric based on AI reward model score to rank all samples from the alignment fine-tuning dataset. We use ranking to prioritise stronger samples over weaker samples by fine-tuning on our Quality Sorted Dataset (QSD), which significantly improves the alignment of T2I models with human preferences and shows over 10x improvement in fine-tuning efficiency. We demonstrate that over 90% of the samples in Pick-a-Pic dataset sends conflicting signals which does more harm than good during RLHF fine-tuning. Finally, we demonstrate through human and AI evaluations that our ranking method improves the performance of state-of-the-art fine-tuning techniques and is preferred by human raters. For brevity, we refer to our approach as DPO-QSD or QSD. Figure 2 shows the generated image outputs from SDXL base, SDXL-DPO checkpoint fine-tuned on full Pick-a-Pic v2 dataset (Kirstain et al., 2023) of approximately 1 million image preference pairs, and SDXL-DPO-QSD fine-tuned on top 50k image preference pairs selected via our method.

## 2 RELATED WORK

The alignment of diffusion models with human preferences has become a critical area of research, especially as these models are being used increasingly to generate content with specific objectives. Alignment of diffusion models to human preferences can largely be categorized into two broad categories - with a reward model and without a reward model. Approaches like DRAFT(Clark et al., 2024), AlignProp(Prabhudesai et al., 2023), ReFl(Xu et al., 2023), and ImageReward(Xu et al., 2023) directly backpropagate the gradients from a differentiable reward model to fine-tune the diffusion model. These approaches work for a finite vocabulary set, but do not generalize well to an open vocabulary set and struggle to optimize for complex reward functions like CLIP score Radford et al. (2021). Different reward models are used to fine-tune the diffusion model based on the end-user task. DPOK(Fan et al., 2023) and DDPO(Black et al., 2023) are Reinforcement Learning based approaches that maximize the score from the reward model over a set of limited prompts which limits the performance of these methods as the number of prompts increases. DOODL(Wallace et al., 2023b) attempts to generate more aesthetically pleasing images by doing iterative improvements to the generation at run-time.

The other set of approaches which do not use an explicit reward model are inspired from the success of direct preference optimization. The recent work of Diffusion-DPO(Wallace et al., 2023a) is able to fine-tune a diffusion model on a dataset of prompts and image pairs by reformulating the loss function. Diffusion-KTO(Li et al., 2024) builds on top of Diffusion-DPO and does not require pairwise preference data, allowing fine-tuning of diffusion models on single image feedback. Additionally, D3PO(Yang et al., 2023) suggest creating its own image pairs from a set of prompts and then using a reward model to identify preferred images. Despite all these advances, these approaches still suffer from noisy pairwise preference datasets and over-optimization.

Most diffusion models (Rombach et al., 2021; Podell et al., 2023; Chen et al., 2023; 2024; Esser et al., 2024) are sometimes trained in two stages, where the first stage involves training on a broad dataset followed by fine-tuning on carefully selected good dataset that is more preferred by humans. These models do full fine-tuning of the diffusion model on a subset of 'good' images which are selected via an AI reward model, usually an aesthetic classifier. Parrot(Lee et al., 2024) uses Pareto-optimal sorting to rank images on multiple reward scores to select the optimal subset. Models like DALLE-3(Shi et al., 2020), SD3(Esser et al., 2024), and CogView(Ding et al., 2021) re-caption existing web-scraped datasets to improve text fidelity. However, these approaches require large amount of resources to caption millions of Images.

## 3 METHOD

### 3.1 BACKGROUND

Diffusion-DPO (Wallace et al., 2023a) considers a setting with a fixed dataset $D = \{(c, x_w, x_l)\}$ where each samples consists of a prompt or caption $c$, a winning image $x_w$, and a losing image $x_l$. The aim is to train a new model $p_\theta$ on these preference pairs, which is more aligned with human preferences compared to the reference model $p_{ref}$. Diffusion-DPO achieves this by completely removing the reward model and reformulating the loss as a function to encourages more denoising at $x_w$ than $x_l$.

However, as we can observe from Figure 1a, human preference is subjective, and this sometimes results in noisy labels. Existing approaches do not try to identify these noisy labels and use the entire dataset for fine-tuning as is. For instance, Diffusion-DPO selects all image preference pairs, excluding only those with ties, without any validation of the preferences.

### 3.2 QUALITY METRIC FOR RANKING PREFERENCE PAIRS

We are now able to capture human preferences from online forums. While all the preferences are made by humans, various factors can affect their judgement. Since they are not fully vetted, the reviewer might have malicious intent, different creative, domain and technical knowledge. Most importantly, preferences are highly subjective. Therefore, we look for pairs that are more aligned with overall preference.

Diffusion-KTO (Li et al., 2024) selects samples where win rate of an image w.r.t all the images it was compared with i.e it selects a pair if the winning image won in all the comparisons of the image and losing image lost in all the comparisons made with it. Though this might be a theoretically sound approach, considering the Pick-a-Pic(Kirstain et al., 2023) dataset, less than 5% of the images were compared more than five times and only 25% of the images were actually compared more than once. These low numbers, combined with the fact that the comparisons were made by random individuals, make this an unreliable metric.

We propose a quality metric for each sample, where a higher score indicates a greater likelihood of the pair being correctly labeled. Through experiments, we demonstrate that fine-tuning with higher-quality pairs leads to improved model performance. However, as lower-quality pairs are introduced, performance begins to decline, supporting the importance of ranking image preference pairs.

Consider any paired preference dataset $D = \{(c^1, x_w^1, x_l^1), (c^2, x_w^2, x_l^2), ...., (c^n, x_w^n, x_l^n)\}$, where each sample consists of a caption $(c)$, a winning image $(x_w)$, and a losing image$(x_l)$. We use an AI reward model trained to model human preferences to get the probability of the winning image to be winning and the losing image to be losing. We use the HPSv2(Wu et al., 2023) model that is trained on a expert-reviewed dataset for human preference to output preference for image given the prompt. This preference value will ranges from 0-1, allowing us to interpret them as the probability of the image being preferred. We refer to this model as $\psi$. Now quality $Q$ of each sample pair can be written as

$$Q(c, x_w, x_l) = \psi(x_w/c) * (1 - \psi(x_l/c)) \tag{1}$$

This can be viewed as probability of pair being correct i.e. probability of the winning image being the winning image and the losing image being the losing image.

In Figure 1b, we see a sharp decrease in quality score for the initial 100k pairs, followed by a gradual decline for the majority of the dataset, and finally, another sharp drop towards the end, where the samples are of the poorest quality. This plot illustrates that the dataset has good samples where the winning image is clearly better, average samples where the preference is more subjective and bad samples where the reward model does not agree with human labels.

# 4 EXPERIMENTS

## 4.1 DATASET

We demonstrate the efficacy of our model on the Pick-a-Pic v2 dataset (Kirstain et al., 2023), which is a crowd sourced dataset. A human reviewer is presented with a caption and a pair of images generated by T2I models like Stable Diffusion 2.1 (Rombach et al., 2021), Dreamlike Photoreal 2.05, and Stable Diffusion XL (Podell et al., 2023) variants. The reviewer selects one of the two presented images as more preferred or marks it as a tie. The dataset contains 1 million rows split into 959.5k rows, 20.5k rows, 20.5k rows of train, validation and test sets respectively. The training set contains approximately 58k distinct captions.

## 4.2 HYPER-PARAMETERS

We run experiments on SD1.5(Rombach et al., 2021) and SDXL(Podell et al., 2023) models. For pairwise preference fine-tuning we use the fine-tuning approach as highlighted in Diffusion-DPO(Wallace et al., 2023a). For both set of experiments we use the ADAMW optimizer. For all SD1.5 and SDXL experiments we use a batch size of 128. All experiments are run on a cluster of 8 NVIDIA 80 GB A100 GPUs. We train at fixed square resolution of 512x512 for SD1.5 and 1024x1024 for SDXL. We train for 1 epoch with a learning rate of $1e^{-4}$ for SD1.5 and $1e^{-5}$ for SDXL. In line with the Diffusion-DPO paper, we use a Beta value of 2000 for SD1.5 and 5000 for SDXL. We do not use any dataset augmentations and keep learning rate constant with no warm-up. For all our experiments we fine-tune using the LoRA approach and use a rank of 64 for both SD1.5 and SDXL.

## 4.3 EVALUATION

To verify the effectiveness of our approach we compare against the state-of-the-art human preference learning approaches like Diffusion-DPO (Wallace et al., 2023a) fine-tuned on the entire training dataset. As we use the LoRA technique, we also fine-tune LoRAs for the state-of-the-art approaches and compare against them. We evaluate all checkpoints on the Pick-a-Pic validation set (Kirstain et al., 2023), which consists of 500 unique prompts. We choose four AI reward models: ImageReward (Xu et al., 2023), Pickscore, HPS-v2 Wu et al. (2023) and Laion aesthetics classifier. ImageReward is the first general-purpose text-to-image human preference reward model, which is trained on a total of 137k pairs of expert comparisons. PickScore is a CLIP-based scoring model with a variant of InstructGPT's (Ouyang et al., 2022) reward model objective. Laion aesthetics classifier is also a CLIP based model with a pretrained MLP that is used to measure the aesthetic quality of an image. We also present scores from HPS-v2 scoring model on the HPS-v2 test set, which consists of 3200 prompts. HPSv2 is a preference prediction model trained on the HPD-v2 dataset. HPS-v2 can be used to compare images generated with the same prompt. Additionally, we perform a user study to compare our approach to the state-of-the-art Diffusion-DPO. Similar to Diffusion-DPO, we employ reviewers to select the preferred generation under three different criteria: Q1 General Preference (Which image do you prefer given the prompt?), Q2 Visual Appeal (prompt not considered) (Which image is more visually appealing?) Q3 Prompt Alignment (Which image better fits the text description?). Five responses are collected for each comparison with majority vote (3+) being considered the collective decision. For the user study, we randomly sample 25 prompts from each of the four sub-sections of the HPS-v2 test set: photos, anime, paintings and concept-art.

# 5 RESULTS

In Figure 3 for SD1.5(Rombach et al., 2021) and Figure 4 for SDXL(Podell et al., 2023), we show that the models fine-tuned using Diffusion-DPO(Wallace et al., 2023a) on our quality sorted dataset (QSD) significantly outperform the baseline models fine-tuned using Diffusion-DPO on randomly sampled data across four key metrics. These results are also presented in Table 1. We also observe a significant improvements in fine-tuning efficiency with our SD1.5 DPO-QSD model and the SDXL DPO-QSD model outperforming the baseline models with just 5.33% of the data. As our fine-tuning data increases, we see a peak in the performance of both models after which the metrics start decreasing or start plateauing. This proves our initial hypothesis that not all fine-tuning pairs are

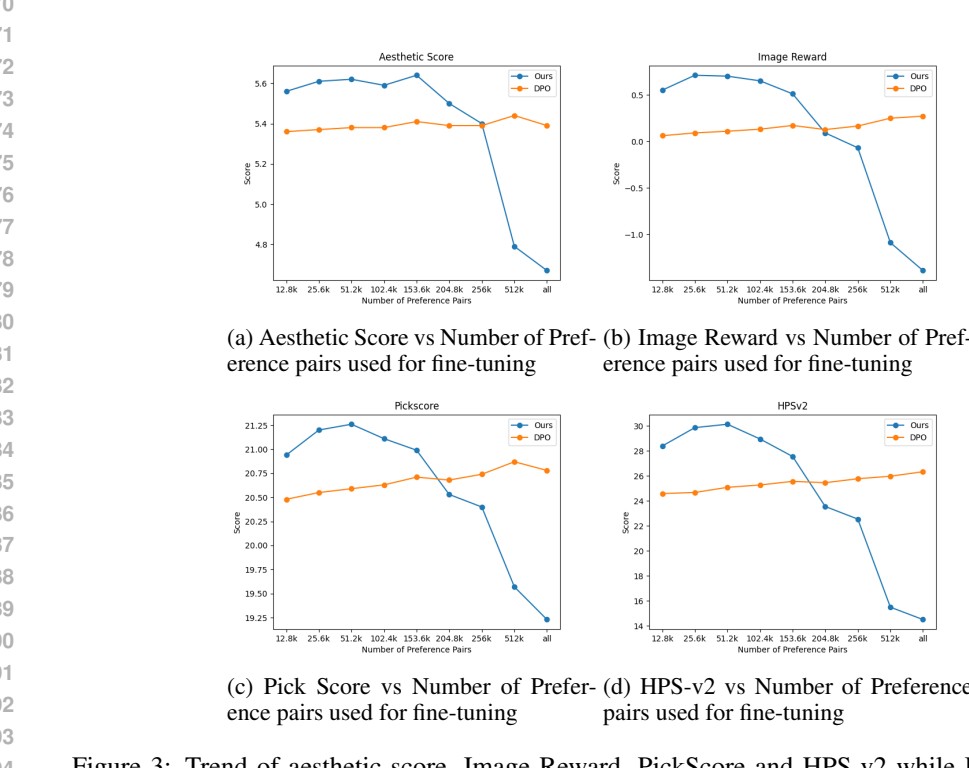

(a) Aesthetic Score vs Number of Pref-
erence pairs used for fine-tuning

(b) Image Reward vs Number of Pref-
erence pairs used for fine-tuning

(c) Pick Score vs Number of Prefer-
ence pairs used for fine-tuning

(d) HPS-v2 vs Number of Preference
pairs used for fine-tuning

Figure 3: Trend of aesthetic score, Image Reward, PickScore and HPS-v2 while Diffusion-DPO
fine-tuning of SD 1.5 on our quality-sorted dataset vs full dataset.

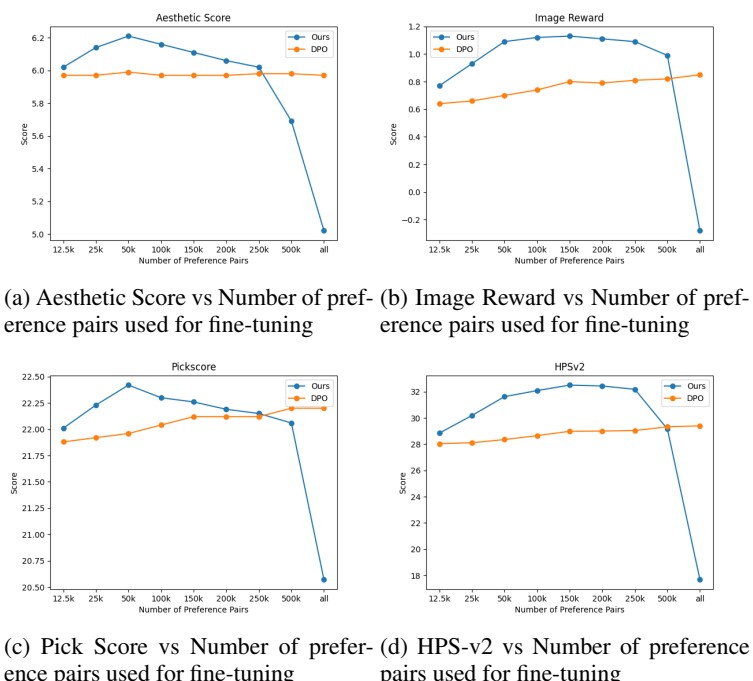

(a) Aesthetic Score vs Number of pref-
erence pairs used for fine-tuning

(b) Image Reward vs Number of pref-
erence pairs used for fine-tuning

(c) Pick Score vs Number of prefer-
ence pairs used for fine-tuning

(d) HPS-v2 vs Number of preference
pairs used for fine-tuning

Figure 4: Trend of aesthetic score, Image Reward, PickScore and HPS-v2 while Diffusion-DPO
fine-tuning of SDXL on our quality-sorted dataset vs full dataset.

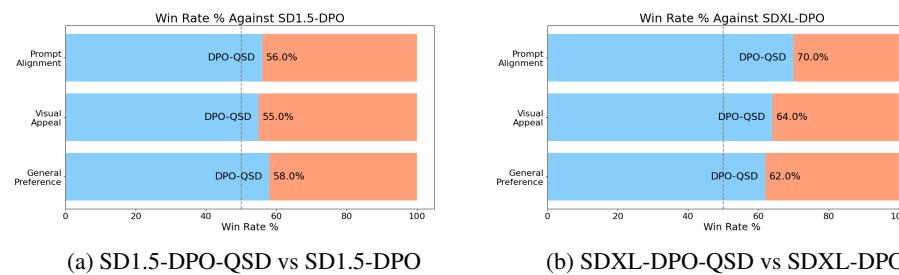

(a) SD1.5-DPO-QSD vs SD1.5-DPO      (b) SDXL-DPO-QSD vs SDXL-DPO

Figure 5: SD1.5 and SDXL QSD models significantly outperform the baseline models in human evaluation.

Table 1: Comparison of our DPO-QSD approach with baseline DPO for SD1.5 and SDXL. With our dataset ranking approach we are able to achieve superior performance over baseline while only using 5.33% of the dataset.

| Method | Aesthetic Score | Image Reward | PickScore | HPSv2 | Samples Used |
|---|---|---|---|---|---|
| SD1.5 DPO | 5.39 | 0.27 | 20.78 | 26.34 | 100% |
| **SD1.5 DPO-QSD** | **5.62** | **0.70** | **21.26** | **30.14** | **5.33%** |
| SDXL DPO | 5.97 | 0.85 | 22.20 | 29.40 | 100% |
| **SDXL DPO-QSD** | **6.21** | **1.09** | **22.42** | **31.62** | **5.33%** |

equal and that some fine-tuning data does more harm than good by sending adverse signals. By using only 5.33% of the Pick-a-Pic dataset we achieve our best models, which vastly outperform the baseline models fine-tuned on the full training dataset. This also proves that over 90% of the preference pairs in Pick-a-Pic v2 (Kirstain et al., 2023) dataset negatively impact training and can be discarded.

Similarly, the user study in Figure 5 shows that our models are preferred by human raters over baseline Diffusion-DPO models. Our SDXL DPO-QSD model is preferred by human annotators 70% of the time in prompt alignment, 64% of the time in visual appeal and 62% of the time in general preference. Similarly, our SD1.5 DPO-QSD model is preferred by human annotators 54% of the time in prompt alignment, 55% of the time in visual appeal and 58% of the time in general preference. We also highlight examples of the high-quality pairs in Figure 7 and low-quality pairs in Figure 6, ranked using our approach.

## 5.1 EFFICACY WITH DIFFERENT FINE-TUNING METHODS

We fine-tune the base model using different fine-tuning methods to show that our QSD is effective in improving performance across different fine-tuning approaches. For all experiments, we fine-tune the baseline on the train dataset with random sampling, while our approach uses the quality sorted dataset. We experiment with the loss function of Diffusion ORPO and loss function defined in SLIC-HF(Zhao et al., 2023). We run this ablation using LoRA approach for SD1.5 with rank 64, a batch size of 128, and a learning rate of 1e-4. For Diffusion-ORPO inspired from ORPO(Hong et al., 2024), we use a learning rate of 1e-3 for baseline model and our model as well. We use the quality metric as described in the methodology section.

For these experiments, we select the best-performing model and present the results in table 2. For comparison, we use four different metrics - Aesthetic Score, Image Reward, PickScore and HPSV2 score. As we can observe, our approach performs considerably better than the baseline across both the methods. Moreover, our approach achieves these results while using only the top 5.33% of the data in case of SLIC-HF and top 10.6% of the data for ORPO, demonstrating over a 10x gain in fine-tuning efficiency. This ablation proves that our pair ranking method improves performance across different fine-tuning paradigms and is not limited to the Diffusion-DPO loss formulation. We

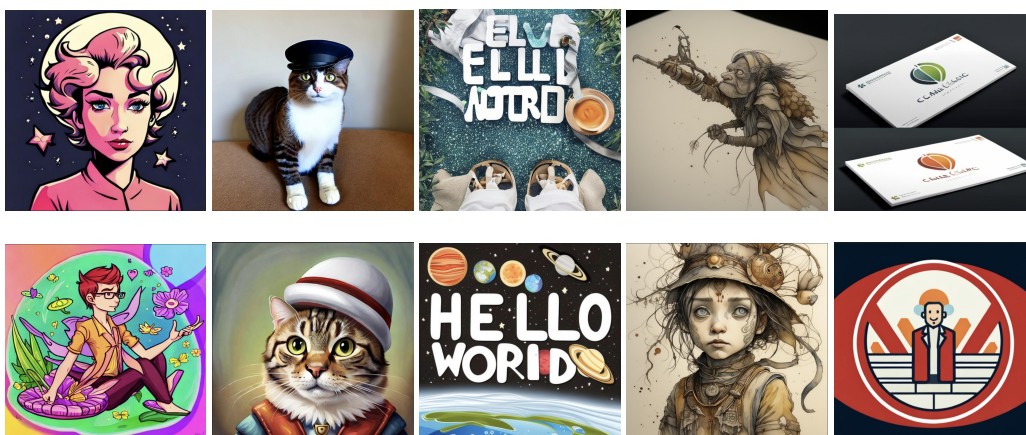

Figure 6: Examples of bad pairs identified by our method. *Top row: Winning Image, Bottom row: losing image*. As can be observed, in these pairs the losing image is better in some quality like aesthetics or prompt adherence over the winning image. Caption from left to right: *(1)a little faery floating in the style of dan hipp, (2) cat wearing a hat, (3) "Hello world" text, space, planets style, (4) face close up woman Jean-Baptiste Monge, watercolour and ink, intricate details, a masterpiece, dynamic backlight, (5) Design a logo for a modern, high-end medical clinic that specializes in personalized, holistic healthcare. The clinic is called "C" and focuses on improving patients' overall well-being through nutrition, exercise, and mental health support. The logo should be simple, sleek, and convey a sense of warmth and approachability while still exuding professionalism and expertise*

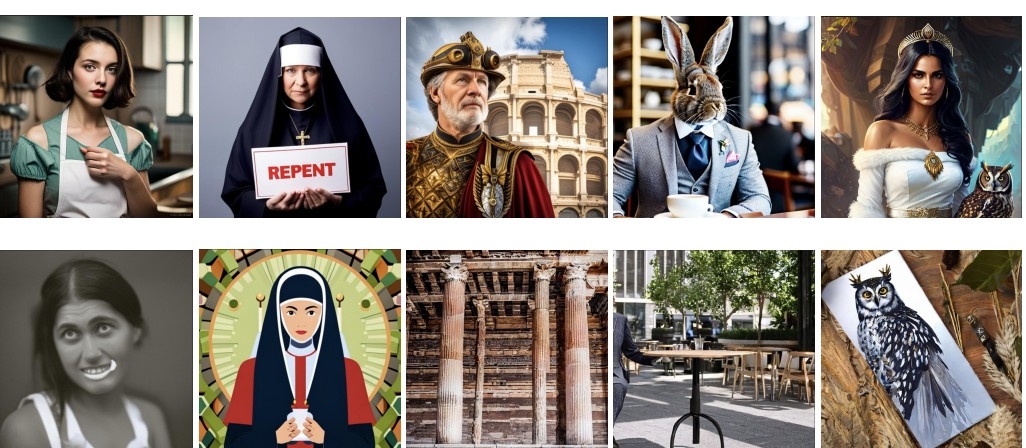

Figure 7: Examples of good pairs ranked best using our method. *Top row: Winning Image, Bottom row: losing image*. The winning images of good samples have better prompt adherence, aesthetic score and are more preferable to humans. Caption from left to right: *(1) A closeup portrait of a playful maid, undercut hair, apron, amazing body, pronounced feminine features, kitchen, freckles, flirting with camera, (2) A nun holding a sign that says repent, (3) Roman emperor, photo, palace background, (4) A rabbit in a 3 piece suit, sitting in a cafe. Hyper Realistic, ultra realistic, 8k, (5)a painting of a woman with an owl on her shoulder, james gurney and andreas rocha, owl princess with crown, also known as artemis or selene, wlop and sakimichan, detaild, portrait character design, falcon, portrait of modern darna, crowned, golden goddess, white witch, by Johannes Helgeson, goddess of travel*

believe that the loss in efficiency for Diffusion-ORPO stems from the inclusion of the mean squared error loss of the winning image in the overall loss function, which dominates the other loss terms

Table 2: Efficacy of our ranking method on different fine-tuning paradigms using Pick-a-Pic dataset. The results prove that our ranking approach gives performance and training efficiency improvement across different fine-tuning approaches.

| Method | Aesthetic Score | Image Reward | PickScore | HPSv2 | Samples Used |
|---|---|---|---|---|---|
| SLIC-HF baseline | 5.45 | 0.33 | 20.93 | 26.71 | 100% |
| **SLIC-HF-QSD** | **5.69** | **0.72** | **21.24** | **29.65** | **5.33%** |
| ORPO baseline | 5.51 | 0.30 | 20.57 | 26.97 | 100% |
| **ORPO-QSD** | **5.60** | **0.60** | **20.80** | **28.25** | **10.6%** |

Table 3: Effect of different scoring models. As we can observe, model trained on just 5.33% of pairs ranked best using HPS-v2 greatly outperform the baseline trained on 100% of the data.

| Method | Aesthetic Score | Image Reward | PickScore | HPSv2 | Samples Used |
|---|---|---|---|---|---|
| Baseline DPO | 5.39 | 0.27 | 20.78 | 26.34 | 100% |
| Image Reward | 5.40 | 0.32 | 20.88 | 26.91 | 100% |
| Laion Aesthetics | **5.80** | 0.49 | 21.09 | 27.30 | 16% |
| PickScore | 5.44 | 0.38 | 21.05 | 27.52 | **5.33%** |
| HPS-v2 | 5.62 | **0.70** | **21.26** | **30.14** | **5.33%** |

## 5.2 EFFECT OF DIFFERENT SCORING MODELS

We test the importance of various scoring models by using different reward models to score each pair of images. We use the loss function defined in the Diffusion-DPO **?** paper as our fine-tuning approach. We run this ablation using LoRA approach for SD1.5 with rank 64, learning rate 1e-4, and a batch size of 128. We keep the quality function constant as $\psi_z(c, x_w) * (1 - \psi_z(c, x_l))$. For this experiment, we try out four different scoring models $\psi_z(c, x_w)$ - HPSv2(Wu et al., 2023), Laion aesthetic score predictor, PickScore(Kirstain et al., 2023) model, and ImageReward(Xu et al., 2023) model. To view theses scoring models as probabilities, we standardized the PickScore and clip the values to +/- 3 which removes the outliers beyond 99% values then shift them to 0-1 by adding 3 and divide with 6. Aesthetic score is divided by 10. Image reward is in range of -3 to 3, we do similar shift as in pickscore. We divide the laion aesthetic score ranging 0-10 by 10

We present the results in Table 3. As we can observe, the model fine-tuned on pairs ranked best using HPS-v2 as the scoring model all other scoring models. ImageReward fails to serve as a good ranking metric for pairs. While the Laion aesthetic predictor shows great improvement in aesthetic score as expected, it fails to show similar improvement across other metrics. HPS-v2 slightly outperforms PickScore and achieves the best results using only 5.33% of the dataset. This ablation reinforces our use of HPS-v2 as a scoring metric.

## 5.3 EFFECT OF LORA RANK

To test the effect of capacity of the LoRA layers and their effect on the model's capability to learn the new information from the dataset, we run experiments with different dimensions of the LoRA layers. Specifically, we want to see how the performance of the model and the fine-tuning efficiency varies with our QSD dataset as we vary the LoRA rank. We run this experiment using SD1.5 as the base model with a learning rate of 1e-4 and a batch size of 128. To this end, we fine-tune with three different LoRA ranks - 32, 64 and 256. For comparison with the baseline, we fine-tune dpo models with the same hyper-parameters and ranks. We present the results in Table 4. As we can observe, we achieve the best results with rank 256 LoRA; however, the improvements over rank 64 are minimal. Therefore, we decide to use rank 64 for our main results. The key observation is that despite the capacity of the LoRA model we get the best fine-tuning efficiency with just 5.33% of the data.

Table 4: Effect of LoRA rank on training efficiency and model performance. Despite different LoRA sizes we get out best model at 5.33% of the data which shows that the best selected pairs are independent of model size.

| Method | Rank | Aesthetic Score | Image Reward | PickScore | HPSv2 | Samples Used |
|---|---|---|---|---|---|---|
| Baseline DPO | 32 | 5.43 | 0.25 | 20.93 | 26.39 | 100% |
| DPO-QSD | 32 | **5.58** | **0.68** | **21.24** | **29.82** | **5.33%** |
| Baseline DPO | 64 | 5.39 | 0.27 | 20.78 | 26.34 | 100% |
| DPO-QSD | 64 | **5.62** | **0.70** | **21.26** | **30.14** | **5.33%** |
| Baseline DPO | 256 | 5.42 | 0.35 | 20.91 | 26.65 | 100% |
| DPO-QSD | 256 | **5.66** | **0.70** | **21.24** | **30.06** | **5.33%** |

## 6 LIMITATIONS AND FUTURE WORK

In this work, our focus was primarily on training a better preference optimized model using a minimal fraction of the dataset. However, we acknowledge that there are limitations to our approach. Our approach relies on a quality metric for selecting training pairs, which could potentially overlook specific edge cases. While we discard the ambiguous preference pairs, future works could find a way to leverage this data as well without any performance loss. Future works could also explore the use of the quality metric as a dynamic threshold, not just for ranking pairs but also for constructing pairwise preference datasets that balance high-quality and nuanced samples.

## 7 CONCLUSION

In this paper, we address the problem of optimal fine-tuning of diffusion models to better align them with human preferences. Unlike previous approaches, we solve this problem by introducing a quality metric that prioritizes high-quality preference pairs and fine-tune in a sorted fashion on this dataset. We demonstrate that our data ranking strategy significantly enhances diffusion model alignment, achieving superior results across multiple AI-based metrics and human evaluators. Our experiments show that models fine-tuned with less than top 10% of the Pick-a-Pick v2 dataset outperform baseline models in both quantitative metrics and human preference evaluations. We run multiple ablations to showcase the effectiveness of our data ranking approach across multiple methods. We validate our initial hypothesis that not all preference pairs contribute equally, and fine-tuning on the entire dataset can be detrimental. By applying our fine-tuning strategy alongside early stopping, one can significantly enhance training efficiency, leading to a more robust and powerful model.

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
