# OpenReview forum: "Effective Text-to-Image Alignment with Quality Aware Pair Ranking"
_ICLR.cc/2025/Conference — Submitted to ICLR 2025_

### Official Review · Reviewer_8QGK · 2024-10-27

**Soundness:** 3
**Presentation:** 2
**Contribution:** 2
**Rating:** 5
**Confidence:** 4

**Summary:**

This paper introduces a sample selection approach specifically designed for the training of text-to-image generation models. Through a set of conducted experiments, the method has shown to be beneficial in improving the training process of such models. The results indicate that the proposed selection technique can contribute to the efficiency and quality of the generated images. Furthermore, the data generated and used in this study has been made available to the community, which may be of use for other researchers working in the text-to-image generation domain.

**Strengths:**

1. The paper addresses a relatively novel topic in the field of image generation.
2. It presents a study that has conducted a series of experiments to explore and validate the proposed approach.

**Weaknesses:**

1. The paper employs the HPSv2 to filter samples in a computational study. The novelty of the approach is quite limited. Additionally, the paper notes the potential bias inherent in using HPSv2 as the sole scoring metric and recommends incorporating a variety of indicators for a more balanced evaluation. The results tabulated in the paper suggest that the HPSv2 metric shows a more pronounced improvement compared to other metrics when contrasted with the baseline model. Thus, the authors emphasize the necessity for a more extensive experimental validation to comprehensively confirm the superiority of the QSD method.
2. This paper presents a study focused on the quality of a model trained on a single dataset, which limits the generalizability of the findings as it does not extend the validation to other datasets. The lack of cross-dataset testing makes it challenging to assert the model’s ability to generalize its quality performance.
3. A recommendation is made to include a comparative experiment where a subset of the full dataset, randomly sampled to match the size of the QSD, is used for benchmarking purposes.
4. Furthermore, the organization and layout of the paper could be improved; for instance, the placement of Figure 3 and Figure 4 appears to be inappropriate, and the design of the tables is somewhat incomplete, which could benefit from a more refined presentation.

**Questions:**

Please refer to weaknesses.

---

> ### Author Response · Authors · 2024-11-21
> **Response to Reviewer 8QGK**
>
> **Weakness: The paper employs the HPSv2 to filter samples in a computational study. The novelty of the approach is quite limited. Additionally, the paper notes the potential bias inherent in using HPSv2 as the sole scoring metric and recommends incorporating a variety of indicators for a more balanced evaluation. The results tabulated in the paper suggest that the HPSv2 metric shows a more pronounced improvement compared to other metrics when contrasted with the baseline model. Thus, the authors emphasize the necessity for a more extensive experimental validation to comprehensively confirm the superiority of the QSD method.**
>
> **Response:** Thank you for this constructive feedback. Our work focuses on demonstrating the practical impact of a quality metric through extensive experimentation. In our study, we carefully assess various fine-tuning methods, LoRA rank configurations, and scoring models to validate the metric’s effectiveness. By running these diverse experiments, we show that despite the simplicity of our approach, our method consistently improves fine-tuning efficiency and enhances overall model performance. We hope this strengthens the contribution of our work as a resource for those prioritizing practical gains in fine-tuning and model optimization.
>
> **Weakness: This paper presents a study focused on the quality of a model trained on a single dataset, which limits the generalizability of the findings as it does not extend the validation to other datasets. The lack of cross-dataset testing makes it challenging to assert the model’s ability to generalize its quality performance.**
>
> **Response:** Thank you for your thoughtful review and for pointing out the importance of cross-dataset validation. At present, the lack of suitable preference datasets in the open-source community poses a challenge in assessing the model's broader applicability.  However, it is important to note that even the other top papers in this field, such as DPO/ KTO/ SPO, all demonstrate their effectiveness primarily on the Pick-a-Pic dataset, as it is currently the only widely available benchmark for such evaluations. Consequently, our study is limited to the most relevant and only available dataset. To address this gap, future work could involve developing a new preference dataset from scratch, utilizing our quality metric to guide the curation process.
>
> **Weakness: A recommendation is made to include a comparative experiment where a subset of the full dataset, randomly sampled to match the size of the QSD, is used for benchmarking purposes.**
>
> **Response:** To address the recommendation, we would like to clarify that we have already conducted a comparative experiment to analyze model performance on both randomly sampled and metric-sorted subsets of the data. In our experiments, we sort the training data by these quality scores and evaluate model checkpoints every 200 steps. This process enables us to iteratively train on progressively larger subsets of the data within a single training run. We’ve evaluated model checkpoints across 4 different metrics at regular intervals and present these results in Figure 3 for SD1.5 and Figure 4 for SDXL. This allows for an understanding of the model's progression across different data quantities. The yellow line represents the baseline model trained with different sized randomly sampled subsets of the training data as shown on the x-axis.
>
> **Weakness: Furthermore, the organization and layout of the paper could be improved; for instance, the placement of Figure 3 and Figure 4 appears to be inappropriate, and the design of the tables is somewhat incomplete, which could benefit from a more refined presentation.**
>
> **Response:** We recognize this issue and have improved the organization and layout of the paper, including repositioning Figures 3 and 4 and enhancing the design of the tables for a more refined presentation.

---

> > ### Comment · Reviewer_8QGK · 2024-11-26
> >
> > Thank you for the author's response, which addressed some of my concerns. Taking into account the opinions of other reviewers, I choose to maintain my original rating.

---

> ### Author Response · Authors · 2024-11-30
> **Response to Reviewer 8QGK**
>
> We value your feedback and have made efforts to address all the concerns you raised.
>
> We demonstrated our papers practical impact through extensive experiments, including testing various fine-tuning methods, LoRA configurations, and scoring models. These experiments highlight consistent improvements in model performance, showcasing the robustness and utility of our approach.
>
> On the generalizability of our findings, we explained the reliance on the Pick-a-Pic dataset, which is a common practice in this field due to the lack of other open-source preference datasets. While this limitation also affects other leading works like DPO, KTO, and SPO, we suggested developing a new preference dataset as an avenue for future work to address this gap.
>
> For your recommendation to include a comparison with randomly sampled subsets, we clarified that such experiments were already conducted and explained them again for your convenience. Figures 3 and 4 illustrate the performance of models trained on random and QSD subsets, enabling direct benchmarking and comparison.
>
> Lastly, regarding the layout and presentation, we improved the organization of the paper by repositioning Figures 3 and 4 and refining the table designs for clarity and readability.
>
>
> That said, we find it challenging to further address your concerns without knowing which specific aspects remain unresolved. Could you please help us understand what would precisely help address the remainder of your concerns? If there are particular areas where you believe additional details, evidence, or explanations are needed, we would be happy to elaborate further. If there are any responses, we provided to other reviewers that you think could be strengthened or elaborated upon, we would be happy to revisit and refine those as well. Your actionable feedback would greatly help us refine our work and ensure we comprehensively address your concerns.

---

### Official Review · Reviewer_EBAc · 2024-10-30

**Soundness:** 2
**Presentation:** 2
**Contribution:** 1
**Rating:** 3
**Confidence:** 4

**Summary:**

This paper introduces a method to enhance text-to-image diffusion models' alignment with human preferences by ranking image pairs with a quality metric. It shows that fine-tuning on just 5.33% of the best-ranked pairs from the Pick-a-Pic dataset outperforms models fine-tuned on the full dataset, demonstrating efficiency and robustness across various methods.

**Strengths:**

1. The paper demonstrates that by using a quality-aware pair ranking method, the alignment of T2I models with human preferences is improved, leading to better quantitative and qualitative results.
2. A key advantage is the dramatic increase in fine-tuning efficiency. The authors show that their models can achieve superior performance using only a small fraction (5.33%) of the full dataset, indicating a substantial reduction in computational resources and time.

**Weaknesses:**

1. Where did the 5.33% setting come from? Why 5.33%?
2. The paper is more of an experimental report, with its main contribution being the application of a quality metric rather than a significant theoretical or methodological innovation.
3. There's a risk that the model may overfit to the Pick-a-Pic dataset, and its generalizability to other datasets or domains is not fully explored.
4. The approach heavily depends on AI reward models, which, if biased or inaccurate, could negatively affect the fine-tuned T2I models' alignment with genuine human preferences.

**Questions:**

See Weaknesses.

---

> ### Author Response · Authors · 2024-11-21
> **Response to Reviewer EBAc**
>
> **Weakness: Where did the 5.33% setting come from? Why 5.33%?**
>
> **Response:** We appreciate the reviewer’s attention to detail and the question regarding the 5.33% setting. This specific threshold was derived through empirical evaluation, where we assessed model checkpoints trained on incrementally larger data sizes, examining performance after every 200 steps of training. We’ve evaluated model checkpoints across 4 different metrics at regular intervals and present these results in Figure 3 for SD1.5 and Figure 4 for SDXL. This allows for an understanding of the model's progression across different data quantities. Through these evaluations, we observed a consistent performance peak around 5.33% of the data, or approximately 51.2k samples out of 960k total samples. Beyond this point, accuracy begins to decline, a pattern that held across various training methods and ablation experiments. This empirical insight guided our selection of this benchmark, allowing us to capture the model’s optimal performance given our dataset.
>
> **Weakness: The paper is more of an experimental report, with its main contribution being the application of a quality metric rather than a significant theoretical or methodological innovation.**
>
> **Response:** Thank you for this constructive feedback. We recognize that our work does not introduce a new theoretical framework; rather, it focuses on demonstrating the practical impact of a quality metric through extensive experimentation. In our study, we carefully assess various fine-tuning methods, LoRA rank configurations, and scoring models to validate the metric’s effectiveness. By running these diverse experiments, we show that despite the simplicity of our approach, our method consistently improves fine-tuning efficiency and enhances overall model performance. We hope this strengthens the contribution of our work as a resource for those prioritizing practical gains in fine-tuning and model optimization.
>
> **Weakness: There's a risk that the model may overfit to the Pick-a-Pic dataset, and its generalizability to other datasets or domains is not fully explored.**
>
> **Response:** We appreciate you for your thoughtful review and for pointing out the importance of cross-dataset validation. At present, the lack of suitable reference datasets in the open-source community poses a challenge in assessing the model's broader applicability.  However, it is important to note that even the other top papers in this field, such as DPO/ KTO/ SPO, all demonstrate their effectiveness primarily on the Pick-a-Pic dataset, as it is currently the only widely available benchmark for such evaluations. Consequently, our study is limited to the most relevant and only available dataset. To address this gap, future work could involve developing a new preference dataset from scratch, utilizing our quality metric to guide the curation process.
>
> **Weakness: The approach heavily depends on AI reward models, which, if biased or inaccurate, could negatively affect the fine-tuned T2I models' alignment with genuine human preferences.**
>
> **Response:** Thank you for pointing out this important concern regarding AI reward model reliance. We agree that reward models can be biased or inaccurate, potentially misaligning models with true human preferences. However, our study specifically highlights that human-annotated labels, while valuable, are also often noisy and deeply subjective
> To address these challenges, we perform a user study, like the approach used in Diffusion-DPO. In this study, we evaluate the images generated by our models across three key aspects: general preference, visual appeal, and prompt alignment. The results show that the images produced by our models are consistently preferred by the majority of human evaluators across all three dimensions, demonstrating the practical effectiveness of our approach.
> Additionally, we conduct an ablation study with different reward models to examine the inherent biases that arise when using reward models as scoring metrics. This analysis reveals that while reward models can indeed vary in their alignment with human preferences, the Human Preference Score (HPS) stands out as the most robust and least prone to biases among the models we tested. HPS not only achieves superior results but also aligns more closely with genuine human preferences, as evidenced by both our ablation and user study findings.

---

> ### Comment · Reviewer_EBAc · 2024-11-23
>
> Since this paper does not propose its own method and is merely a study and ablation of existing methods, I will maintain my current score.

---

> > ### Author Response · Authors · 2024-11-24
> > **Reply to Reviewer EBAc**
> >
> > Thank you for your question. We would like to assert that our work is not merely an ablation of previous works but includes a novel contribution in the form of a quality metric for paired preference data. To the best of our knowledge, this is the first attempt to study preference pairs and their impact on image-based preference optimization. The motivation behind our approach is that we hypothesize that the samples where the winning and losing images have similar or inverse AI preference ratings negatively impact the model
> > during pairwise preference fine-tuning by sending conflicting signals. We explicitly describe the problem with current paired preference datasets and explain in visually in Figure 1 and Figure 6 and talk about it more in our Introduction.
> >
> > Specifically, our method evaluates the quality of each sample pair using a reward model trained on expert-reviewed data for human preferences. The quality metric, as described in our methods section, allows us to rank preference pairs based on their reliability. The metric is designed to measure the likelihood of a preference pair being beneficial for preference optimization. Given a dataset containing captions, winning images, and losing images, we calculate the quality score of a pair as the product of the winning image's preference score (conditioned on the caption) and the complement of the losing image's preference score. This score quantifies the confidence in the correctness of the pair and enables us to rank and filter pairs effectively, improving fine-tuning outcomes across multiple fine-tuning methods like Diffusion-DPO, Diffusion-ORPO and SLIC-HF (Table 1 and Table 2). We conduct human evaluations to show that our methods outperform existing baselines, while providing an over 90% reduction in training time.
> >
> > When we refer to a **"theoretical framework"**, we specifically mean that our work provides **ranking formulation, quantitative evidence and rigorous empirical validation ... as opposed to making conclusions simply from theoretical proof.** Our proposed method explores empirical results for understanding and advancing modern practices for efficiency in preference optimization. We also show that our approach works across different LoRA ranks (Table 4) and model architectures (SD1.5 and SDXL), effectively proving that our approach is robust to model architectures and size. By addressing limitations in existing methods and offering a robust, validated solution, our work establishes a meaningful contribution to preference optimization. We hope this clarifies the originality and depth of our study. We kindly request you to reconsider your assessment, as we believe our work provides a novel contribution that has been shown to work effectively across various evaluations.

---

### Official Review · Reviewer_ZPvq · 2024-11-02

**Soundness:** 3
**Presentation:** 2
**Contribution:** 2
**Rating:** 5
**Confidence:** 4

**Summary:**

This paper presents a data curation method for DPO based preference optimization for text-to-image models. The main idea is to decide how useful a certain preference is using the proposed quality metric. The proposed quality metric is basically a product of the score for the winning sample, and the inverse (1-x) for the losing sample. Performing LoRA fine tuning with Diffusion-DPO (or its variants) on both SD1.5 and SDXL improves metrics on several evaluation setups (human preference reward model evaluations and user studies).

**Strengths:**

The biggest strength of the paper in my view is that it starts the discussion on data quality for preference datasets of T2I models. While open-source datasets are not widely available, filtering existing datasets to get much better data would be essential to improve the results.

In that context, the results obtained in the paper are quite compelling. The quality metric proposed here is quite simple/straightforward, yet provides consistent gains across different DPO strategies (e.g. ORPO etc.)

**Weaknesses:**

A major "weakness" that can be pointed to this paper is that the contribution is somewhat basic and limited -> i.e introduce a quality metric to rank pairs and select the top ranked papers to perform Diffusion-DPO with LoRA fine-tuning. To that extent, there's limited contribution, and I would love to have seen more insights for curating future preference datasets.

I would have also loved to have seen some more insight into whether the images are an issue, or are the labels wrongly assigned by the annotators? For instance, if we were to relabel the Pick-a-Pic dataset using HPSv2, how does the resulting DPO process look like (the kinds of experiments in Fig.6 and Tab. 2 in the Diffusion-DPO paper). There's also one qualitative figure showing "good pairs" and "bad pairs", but something more concrete/quantitative would also provide a lot more support.

In general, I think the paper is touching upon a crucial aspect of preference datasets for (Diffusion) DPO, and is valuable, but I would really like to see some more analyses and insights before confidently recommending acceptance.

**Questions:**

I am mostly confused by the results in Tab. 3 and would like the authors to clarify the details. The first question is why do different methods have such differing % of samples used? For HPSv2 and PickScore it's 5.33% while for Aesthetic predictor it's 16% and with ImageReward it's all the samples (i.e no filtering)? Couldn't the same number of samples be used for all the scoring methods to keep it more balanced? Especially when no filtering is applied (i.e the ImageReward case), I am not sure what value is being added by the scoring method at all?

---

> ### Author Response · Authors · 2024-11-21
> **Response to Reviewer ZPvq part 1**
>
> **Weakness: A major "weakness" that can be pointed to this paper is that the contribution is somewhat basic and limited -> i.e introduce a quality metric to rank pairs and select the top ranked papers to perform Diffusion-DPO with LoRA fine-tuning. To that extent, there's limited contribution, and I would love to have seen more insights for curating future preference datasets.**
>
> **Response:** Thank you for this valuable feedback. We understand that our contribution may appear straightforward, as it primarily centers on the application of a quality metric to rank pairs and optimize Diffusion-DPO with LoRA fine-tuning. While we agree that this approach is relatively direct, it was motivated by the current scarcity of open-source preference datasets, which limits our ability to extend this work to broader insights in preference data curation.
>
> To address this gap, future work could involve developing a new preference dataset from scratch, utilizing our quality metric to guide the curation process. We believe this effort will lay the groundwork for future research and provide a robust resource for others looking to refine and evaluate preference models. We appreciate your interest in seeing more insight into preference dataset curation and look forward to sharing these developments in future work.
>
> **Weakness: I would have also loved to have seen some more insight into whether the images are an issue, or are the labels wrongly assigned by the annotators? For instance, if we were to relabel the Pick-a-Pic dataset using HPSv2, how does the resulting DPO process look like (the kinds of experiments in Fig.6 and Tab. 2 in the Diffusion-DPO paper). There's also one qualitative figure showing "good pairs" and "bad pairs", but something more concrete/quantitative would also provide a lot more support.**
>
> **Response:** Thank you for your insightful comment. To further investigate whether the issues arise from the images themselves, or incorrect labels assigned by the annotators, we conducted an experiment where we relabeled a portion of the Pick-a-Pic dataset using our HPSv2 model. We train a lora for SD-1.5 using the hyperparameters mentioned in our paper. For our dataset, we took the last 100k samples from Pick-a-Pic, which were sorted as the worst by our quality score and performed two separate training runs: one using the original labels via human preferences, and another where we flipped the labels (i.e., the winning image became the losing image, and vice versa). The results of these experiments are shown in the table below:
> | Number of Samples | Aesthetic score | IR | Pickscore | Hpsv2 | Aesthetic score (flipped) | IR (flipped) | Pickscore (flipped) | Hpsv2 (flipped) |
> | ---- | ---- | ---- | ---- | ---- | ---- | ---- | ---- | ---- |
> | 12.5k | 5.18 | -0.42 | 20.10 | 20.76 | 5.36 | 0.07 | 20.42 | 25.39 |
> | 25k | 5.07 | -0.64 | 19.89 | 18.65 | 5.33 | 0.18 | 20.38 | 25.78 |
> | 50k | 4.86 | -1.10 | 19.54 | 14.93 | 5.35 | 0.16 | 20.38 | 25.69 |
> | 100k | 4.80 | -1.25 | 19.42 | 13.72 | 5.29 | 0.20 | 20.33 | 25.41 |
>
> As we can see, the model trained with the flipped labels generally performs better than the model trained with the original annotations. However, even with the flipped labels, performance still decreases as training progresses. This indicates that the poor performance of these pairs is due to a combination of factors: incorrect annotations and inherent issues with the image quality. Both factors are effectively captured by our quality score. This experiment provides more concrete evidence that both annotation errors and image quality contribute to the performance degradation in training.

---

> ### Author Response · Authors · 2024-11-21
> **Response to Reviewer ZPvq part 2**
>
> **Question: I am mostly confused by the results in Tab. 3 and would like the authors to clarify the details. The first question is why do different methods have such differing % of samples used? For HPSv2 and PickScore it's 5.33% while for Aesthetic predictor it's 16% and with ImageReward it's all the samples (i.e no filtering)? Couldn't the same number of samples be used for all the scoring methods to keep it more balanced? Especially when no filtering is applied (i.e the ImageReward case), I am not sure what value is being added by the scoring method at all?**
>
> **Response:** Table 3 supports our choice of using HPSv2 as the reward scoring model for the quality metric. In our experiments, we sorted the training data by quality scores generated by different reward models and evaluated model checkpoints every 200 steps. We report the metrics, and the percentage of the dataset used for the best checkpoint of each method. Image Reward-based QSD showed minimal improvement over the baseline, with the best model only towards the end of training, utilizing the entire 100% dataset. Aesthetic Score-based QSD resulted in a better-performing model using 16% of the dataset and excelled in the aesthetic score, as expected. PickScore-based QSD achieved superior performance using just 5.33% of the dataset. HPSv2-based QSD delivered the best results across all performance metrics except for the aesthetic score using just 5.33% of dataset.
>
> Choosing the same threshold of 5.33% of the dataset, using reward model other than HPSv2 would result in lower scores than those presented in the Table 3.

---

### Official Review · Reviewer_idWG · 2024-11-04

**Soundness:** 2
**Presentation:** 2
**Contribution:** 2
**Rating:** 3
**Confidence:** 4

**Summary:**

This paper proposes a new approach to improve text-to-image alignment in diffusion models by introducing a quality-aware pair ranking (QSD) mechanism. The key idea is to rank image preference pairs based on an AI-generated quality metric, allowing for more effective fine-tuning with higher-quality samples. Experiments indicate that using only the top-ranked 5.33% of preference pairs results in significant performance improvements, both quantitatively and qualitatively, over models trained with the full dataset. The authors suggest that this ranking approach enables more efficient model alignment with human preferences.

**Strengths:**

1. The proposed approach is straightforward and does not require complex architectural changes, making it accessible for integration with existing systems.
2. The QSD ranking approach demonstrates significant improvement in both alignment and training efficiency, as evidenced by quantitative and qualitative metrics.

**Weaknesses:**

1. My main concern is that this method relies heavily on clear preference pairs, or “easy samples.” While this strategy effectively avoids the noise introduced by pairs with similar quality scores in the dataset, it is highly likely to lead to overfitting on easily distinguishable image pairs, thereby limiting the model’s generalization ability in practical applications. This issue has been discussed in many papers [1, 2, 3]. Specifically:
a. All experiments and evaluations are conducted on the Pick-a-Pic dataset, raising concerns about the model's generalizability.
b. From Figures 6 and 7, it is evident that QSD-selected pairs tend to have clear differences in content but may lack nuanced human preference subtleties. Ignoring this data could prevent the model from learning finer distinctions in preference. It is recommended that the authors validate QSD’s effectiveness on a more granular dataset.
c. Real-world applications require models to handle various complex and ambiguous inputs. By entirely excluding such pairs from training, the model may perform poorly when encountering similarly ambiguous inputs in practical settings.
2. There are indications of noisy data in the Pick-a-Pic dataset. For example, in Figure 6 (row 1, column 3), a winning image does not seem to align well with the prompt, raising questions about data quality. It would be beneficial to conduct an ablation study where QSD filters out low-quality pairs to assess its effectiveness in reducing noise for DPO training.
3. The QSD method relies on filtering existing datasets of image preference pairs, which are costly to construct. This limitation restricts the method’s practical applicability. The authors are encouraged to explore automated pipelines to create high-quality preference data at scale.
4. More detailed explanations of how QSD ranks pairs and why specific thresholds (such as the top 5.33%) were chosen would improve readability and comprehension.
5. It would be beneficial to include a discussion of the limitations of the proposed approach and potential directions for future development.
[1] Contrastive learning with hard negative samples.
[2] Smaug: Fixing failure modes of preference optimisation with dpo-positive
[3] Mixed Preference Optimization: Reinforcement Learning with Data Selection and Better Reference Model

**Questions:**

1. Why are Figures 5(a) and 5(b) identical? Please verify the caption in Figure 5(a) which reads “SD1.5-DPO vs SD1.5-DPO.”
2. In Figure 6, row 1, column 3, the ground truth (GT) label for the winning image seems misaligned with the prompt. Instead of focusing solely on high-quality pairs, would it be more beneficial to filter out noisy data to enhance DPO training?

---

> ### Author Response · Authors · 2024-11-21
> **Response to Reviewer idWG Part 1**
>
> **Weakness: My main concern is that this method relies heavily on clear preference pairs, or “easy samples.” While this strategy effectively avoids the noise introduced by pairs with similar quality scores in the dataset, it is highly likely to lead to overfitting on easily distinguishable image pairs, thereby limiting the model’s generalization ability in practical applications. This issue has been discussed in many papers [1, 2, 3]. Specifically: a. All experiments and evaluations are conducted on the Pick-a-Pic dataset, raising concerns about the model's generalizability. b. From Figures 6 and 7, it is evident that QSD-selected pairs tend to have clear differences in content but may lack nuanced human preference subtleties. Ignoring this data could prevent the model from learning finer distinctions in preference. It is recommended that the authors validate QSD’s effectiveness on a more granular dataset. c. Real-world applications require models to handle various complex and ambiguous inputs. By entirely excluding such pairs from training, the model may perform poorly when encountering similarly ambiguous inputs in practical settings.**
>
> **Response:**
>
> a. We appreciate your feedback and fully understand the concerns. At present, the lack of suitable preference datasets in the open-source community poses a challenge in assessing the model's broader applicability.  Notable preference optimization papers (DPO, SPO, KTO) demonstrate their effectiveness exclusively on pick-a-pic dataset, as it is currently the only widely available benchmark for such evaluations.
>
> b. We respectfully disagree with the characterization that QSD-selected pairs are "easy samples". The samples we select are not necessarily easy for DPO to learn from; instead, they are better samples, specifically chosen to enhance the model's ability to learn effectively. Our approach focuses on selecting training pairs that provide clear and actionable preference signals.
> Ambiguous pairs, on the other hand, often involve nuanced human preference subtleties that are challenging even for humans to discern without domain knowledge or keen observation. These subtleties make the training pairs ambiguous, thereby making it very difficult for the diffusion model to learn from them. Training on such pairs does not provide meaningful signals for optimization and can negatively impact model performance.
> We demonstrate this in Figures 3 and 4, where adding ambiguous pairs—those with subtle differences in human preferences—leads to worse performance across evaluation metrics. By discarding such pairs and focusing on better-quality training data, we avoid introducing noise into the training process, as supported by our extensive evaluations on the Pick-a-Pic validation set and through human evaluations
>
> c. Thank you for your insightful comment. Pick-a-Pic is designed as a diverse dataset, capturing prompts directly from real users in a web application. This approach ensures that it reflects a wide range of real-world scenarios.
> To further address your concern, we perform extensive evaluations using trained reward models and human evaluations. These evaluations demonstrate that our approach generalizes well to various complex and ambiguous scenarios. Moreover, we found that the model trained on our selected training pairs significantly outperforms models that was trained using the excluded training pairs as well. This highlights the importance of our data selection strategy, which focuses on high-quality, informative training examples for robust model performance.
> That said, we acknowledge that domain-specific use cases, such as medical image generation, would indeed require specialized datasets and expert preferences, which are beyond the scope of our dataset.

---

> ### Author Response · Authors · 2024-11-21
> **Response to Reviewer idWG Part 2**
>
> **Weakness and Question: There are indications of noisy data in the Pick-a-Pic dataset. For example, in Figure 6 (row 1, column 3), a winning image does not seem to align well with the prompt, raising questions about data quality. It would be beneficial to conduct an ablation study where QSD filters out low-quality pairs to assess its effectiveness in reducing noise for DPO training.**
>
> **Response:** We acknowledge your concerns regarding noisy data and filtering low quality pairs. In Figure 6, we explicitly showcase examples of low-quality image pairs, where the human preference is not ideal. The GT captions are correctly assigned. However, the appearance of misalignment stems from inaccuracies in the human preference ranking process when this pair was labelled. In this example, the losing image is clearly better, which demonstrates why such pairs negatively impact model performance when included in training.
>
> We would like to emphasize that our method is designed to address exactly this issue. By evaluating at regular intervals during training using our quality-based ranking approach (QSD), we identify and filter out these low-quality pairs. This approach enables us to train on multiple subsets of the data in a single training run.  As shown in Figures 3 and 4, our best-performing models use just 5.33% of the dataset, effectively removing noisy data while retaining high-quality pairs. As we add more paired data beyond this, the evaluation metrics start to plateau or fall.
> We hope this explanation provides clarity on both the intent behind Figure 6 and the effectiveness of our ranking-based filtering methodology in removing noisy data from training. We appreciate the reviewer's suggestion and believe that our approach already incorporates the principle of filtering out low-quality data to improve model performance.
>
> **Weakness: The QSD method relies on filtering existing datasets of image preference pairs, which are costly to construct. This limitation restricts the method’s practical applicability. The authors are encouraged to explore automated pipelines to create high-quality preference data at scale.**
>
> **Response:** We appreciate your interest in seeing more insight into preference dataset curation and look forward to sharing these developments in future work.
>
> **Weakness: More detailed explanations of how QSD ranks pairs and why specific thresholds (such as the top 5.33%) were chosen would improve readability and comprehension.**
>
> **Response:** Thank you for highlighting this. We are happy to provide further details on how QSD ranks pairs and the rationale behind our choice of thresholds.
> QSD ranks pairs by estimating the importance of each pair for fine-tuning, leveraging the HPSv2 reward score. The Quality score is treated as the probability of a pair being important for fine-tuning. For a given pair, the Quality score is calculated as the product of  the winning image’s reward score and 1 – losing image’s reward score.
> In our experiments, we sort the training data by these quality scores and evaluate model checkpoints every 200 steps. This process enables us to iteratively train on progressively larger subsets of the data within a single training run. Through this approach, we consistently found that the best-performing checkpoint corresponds to 5.33% of the dataset—equivalent to 51.2k pairs out of approximately 960k pairs. The threshold of 5.33% was determined empirically as it marked the point where evaluation metrics reached their peak. We chose to present this threshold as a percentage to highlight the significant dataset reduction achieved without compromising model performance. We observe this number to be consistent across multiple lora-ranks and multiple model sizes (SD1.5 and SDXL).
>
> **Weakness: It would be beneficial to include a discussion of the limitations of the proposed approach and potential directions for future development. [1] Contrastive learning with hard negative samples. [2] Smaug: Fixing failure modes of preference optimisation with dpo-positive [3] Mixed Preference Optimization: Reinforcement Learning with Data Selection and Better Reference Model**
>
> **Response:** Thank you for your valuable feedback on the limitations and for pointing out relevant works that can help further refine and expand our approach in future works. We have incorporated a section on limitations and future directions to address these points explicitly in our updated manuscript.
>
> **Question: Why are Figures 5(a) and 5(b) identical? Please verify the caption in Figure 5(a) which reads “SD1.5-DPO vs SD1.5-DPO."**
>
> **Response:** We sincerely apologize for the referencing same image for 5(a) and 5(b). This was an oversight on our part, and we deeply regret any confusion it may have caused. We take full responsibility for the error and understand the importance of maintaining accuracy and consistency in our work. We have corrected the issue in the revised manuscript.

---

> > ### Comment · Reviewer_idWG · 2024-11-27
> >
> > I am grateful for the author's response, which tackled some of my questions.  However, after weighing in the perspectives of fellow reviewers, I have decided to retain my initial rating.

---

> > > ### Author Response · Authors · 2024-11-30
> > > **Response to Reviewer idWG**
> > >
> > > Thank you for taking the time to consider our responses and we're glad to able to address some of your questions.
> > > To summarize, we have provided clarifications on several points, including:
> > >
> > > 1. Generalization of the Model: We acknowledge the potential overfitting risk when relying on "easy samples" and have explained that the QSD method specifically selects informative pairs that help avoid ambiguous or noisy data, which would hinder learning. Additionally, we addressed the generalization capability of our approach based on the Pick-a-Pic dataset, noting that the diversity in this dataset reflects a wide range of real-world scenarios and our extensive evaluations.
> > > 2. Handling Noisy Data: In response to concerns regarding noisy data in the Pick-a-Pic dataset, we have explained our approach for filtering low-quality pairs using the QSD method, demonstrating how our method minimizes noise during training and enhances model performance. We also provided examples from our experiments that show the effectiveness of this filtering process.
> > > 3. Practical Applicability: We addressed the practical limitations of constructing preference datasets, acknowledging that such datasets can be costly and time-consuming to curate. We also expressed our interest in exploring automated pipelines for generating high-quality preference data at scale in future work.
> > > 4. Ranking and Thresholding: We clarified the details of how QSD ranks pairs using the HPSv2 reward score, and explained that the threshold of 5.33% was determined empirically through evaluations of model performance.
> > > 5. Figures and Technical Issues: We also acknowledged and corrected the error regarding the identical Figures 5(a) and 5(b), and apologize for any confusion it may have caused.
> > >
> > > That said, we find it challenging to further address your concerns without knowing which specific aspects remain unresolved. Could you please help us understand what would precisely help address the remainder of your concerns? If there are particular areas where you believe additional details, evidence, or explanations are needed, we would be happy to elaborate further. If there are any responses, we provided to other reviewers that you think could be strengthened or elaborated upon, we would be happy to revisit and refine those as well. Your actionable feedback would greatly help us refine our work and ensure we comprehensively address your concerns.

---

### Meta-Review · Area_Chair_1iwn · 2024-12-20

**Metareview:**

This paper presents a quality aware pair ranking method (based on HPSv2) to filter out bad quality pairs in Pick-a-Pic data sets, and shows the filtered pairwise preference data can achieve better diffusion model finetuning. The direction of improving the quality of human preference data is interesting and the major strength. However, all reviewers argued about the technical novelty/contribution of the proposed method, which is quite straightforward. Moreover, experiments are only conducted on Pick-a-pic and hence is not convincing enough. So my recommendation is "Reject".

**Additional Comments On Reviewer Discussion:**

Besides the major weaknesses of insufficient technical contribution and limited experiments on one Pick-a-Pic data set,
reviewers also asked questions for example 1) whether the proposed approach will have a bias to select easy pairs 2)  where does the setting of top 5.33% selected data come from? 3) whether the quality issues some from the images or the labels, as well as some questions related to technical/experiment details.

The authors provide some rebuttal and resolved part of these issues. However, I feel the major weaknesses remains after the rebuttal.

---

### Decision · Program_Chairs · 2025-01-22

Reject